# Comparative Analysis of PSA Density and an MRI-Based Predictive Model to Improve the Selection of Candidates for Prostate Biopsy

**DOI:** 10.3390/cancers14102374

**Published:** 2022-05-11

**Authors:** Juan Morote, Angel Borque-Fernando, Marina Triquell, Anna Celma, Lucas Regis, Richard Mast, Inés M. de Torres, María E. Semidey, José M. Abascal, Pol Servian, Anna Santamaría, Jacques Planas, Luis M. Esteban, Enrique Trilla

**Affiliations:** 1Department of Urology, Vall d’Hebron Hospital, 08035 Barcelona, Spain; mtriquell@vhebron.net (M.T.); acelma@vhebron.net (A.C.); lregis@vhebron.net (L.R.); jplanas@vhebron.net (J.P.); etrilla@vhebron.net (E.T.); 2Department of Surgery, Universitat Autònoma de Barcelona, 08193 Barcelona, Spain; 3Department of Urology, Hospital Miguel Servet, IIS-Aragon, 50009 Zaragoza, Spain; aborque@comz.org; 4Department of Radiology, Vall d’Hebron Hospital, 08035 Barcelona, Spain; rmast@vhebron.net; 5Department of Pathology, Vall d´Hebron Hospital, 08035 Barcelona, Spain; itorres@vhebron.net (I.M.d.T.); mesemidey@vhebron.net (M.E.S.); 6Department of Morphological Sciences, Universitat Autònoma de Barcelona, 08193 Barcelona, Spain; 7Department of Urology, Parc de Salut Mar, Universitat Pompeu Fabra, 08003 Barcelona, Spain; jmabascalj@gmail.com; 8Department of Urology, Hospital Germans Trias i Pujol, 08035 Badalona, Spain; pservian.germantrias@gencat.cat; 9Vall d´Hebron Research Institute, 08035 Barcelona, Spain; anna.santamaria@vhir.org; 10Department of Applied Mathematics, Escuela Universitaria Politécnica La Almunia, Universidad de Zaragoza, 50100 Zaragoza, Spain; lmeste@unizar.es

**Keywords:** prostate-specific antigen density, predictive model, clinically significant prostate cancer

## Abstract

**Simple Summary:**

Magnetic resonance imaging (MRI)-associated prostate-specific antigen density (mPSAD) and MRI predictive models have been proposed for improving the selection of candidates for prostate biopsy among men with suspected prostate cancer (PCa). While the calculation of mPSAD only requires a simple division, the individual risk assessment of PCa using the available risk calculators is also a swift process. We aim to compare the clinical usefulness of mPSAD and an MRI predictive model that utilises the same predictors as the recently developed and externally validated Barcelona MRI predictive model (MRI-PMbdex).

**Abstract:**

This study is a head-to-head comparison between mPSAD and MRI-PMbdex. The MRI-PMbdex was created from 2432 men with suspected PCa; this cohort comprised the development and external validation cohorts of the Barcelona MRI predictive model. Pre-biopsy 3-Tesla multiparametric MRI (mpMRI) and 2 to 4-core transrectal ultrasound (TRUS)-guided biopsies for suspicious lesions and/or 12-core TRUS systematic biopsies were scheduled. Clinically significant PCa (csPCa), defined as Gleason-based Grade Group 2 or higher, was detected in 934 men (38.4%). The area under the curve was 0.893 (95% confidence interval [CI]: 0.880–0.906) for MRI-PMbdex and 0.764 (95% CI: 0.774–0.783) for mPSAD, with *p* < 0.001. MRI-PMbdex showed net benefit over biopsy in all men when the probability of csPCa was greater than 2%, while mPSAD did the same when the probability of csPCa was greater than 18%. Thresholds of 13.5% for MRI-PMbdex and 0.628 ng/mL^2^ for mPSAD had 95% sensitivity for csPCa and presented 51.1% specificity for MRI-PMbdex and 19.6% specificity for mPSAD, with *p* < 0.001. MRI-PMbdex exhibited net benefit over mPSAD in men with prostate imaging report and data system (PI-RADS) <4, while neither exhibited any benefit in men with PI-RADS 5. Hence, we can conclude that MRI-PMbdex is more accurate than mPSAD for the proper selection of candidates for prostate biopsy among men with suspected PCa, with the exception of men with a PI-RAD S 5 score, for whom neither tool exhibited clinical guidance to determine the need for biopsy.

## 1. Introduction

Early detection of clinically significant prostate cancer (csPCa) decreases the specific mortality of PCa [1]. The classic approach of PCa detection based on systematic biopsies after PCa suspicion, has been disapproved due to the high rates of unnecessary biopsies and the over detection of insignificant tumours (iPCa) [2,3,4,5]. While PCa suspicion continues to be based on serum prostate-specific antigen (PSA) elevation or abnormal digital rectal examination (DRE), the detection of csPCa has improved since the introduction of multiparametric magnetic resonance imaging (mpMRI) and guided biopsies [6,7]. Nevertheless, there remain uncertain scenarios in which a high rate of unnecessary biopsies and over detection of iPCa occur [8,9]. The current high negative predictive value of mpMRI enables clinicians to avoid systematic biopsies in men with prostate imaging report and data system (PI-RADS) < 3; however, this approach may fail to detect 5–20% of existing csPCa in men with a negative mpMRI [10,11,12]. Contrarily, clinicians usually recommend prostate biopsy for men with PI-RADS > 3, as csPCa is usually detected approximately at a rate of 50–60% for men with PI-RADS 4, and up to 95% for men with PI-RADS 5 [13,14,15]. PI-RADS 3 remains the most uncertain scenario, in which csPCa usually does not reach 20% and in which more than 50% of the biopsied lesions reflect iPCa [9,15]. Prostate-specific antigen density (PSAD), MRI-based predictive models (MRI-PM), and modern markers are currently recommended for improving the selection of candidates for prostate biopsy [6,7,16].

PSAD is a classic tool that improves the specificity of serum PSA [17]. However, PSAD can be strengthened by pre-biopsy MRI (mPSAD), which yields a more accurate prostate volume at no additional cost [18]. mPSAD has been analysed in overall populations of men to enhance the detection of csPCa in relation to the PI-RADS score [19,20,21,22]. MRI-PMs are also effective tools. However, few MRI-PMs have been developed from the latest versions of PI-RADS; furthermore, external validation should be performed before their use. Easily accessible risk calculators (RCs) are essential for avoiding nomograms, which are cumbersome and time-consuming [23,24,25,26,27,28,29,30,31,32,33,34,35,36,37]. Recently, the Barcelona MRI-PM was developed from the following independent predictors: PI-RADS score v.2.0, age, serum PSA, DRE, PCa family history, type of biopsy (initial vs. repeat), and MRI-derived prostate volume. External validation was carried out in the same metropolitan area, and a web-RC is freely available at https://mripcaprediction.shinyapps.io/MRIPCaPrediction/ (accessed on 5 March 2022). For the first time, the performance of an MRI-PM has been analysed with regard to PI-RADS categories, showing a net benefit over biopsy for all men in each PI-RADS < 4. The designed RC also incorporates the novel option of selecting the csPCa probability threshold [38], which may facilitate further external validation [15] and improve the model’s performance in specific PI-RADS categories [39]. Furthermore, modern markers can also be helpful; however, they are expensive and require the procurement of new samples of blood or urine, usually after prostate massage [40]. Four-Kallikrein test (4K) [41,42,43,44,45], the Prostate Health Index (PHI) [46,47,48], SelectMDx [49,50,51,52], Proclarix [52,53], and the combination of multiple markers have improved the prognostic performance of mpMRI [30].

This study aims to compare the clinical usefulness of mPSAD and MRI-PMbdex for the proper selection of candidates for prostate biopsy in a sizable, multicentre population of men with suspected PCa, according to PI-RADS categories.

## 2. Materials and Methods

### 2.1. Design, Setting and Participants

A head-to-head comparison of mPSAD and MRI-PMbdex was designed from a multicentre trial conducted in the metropolitan area of Barcelona. The trial involved 2432 men with suspected PCa due to a serum PSA > 3.0 ng/mL or an abnormal DRE. MRI-PMbdex was created from the development cohort of 1486 men studied at Vall d’Hebron Hospital (VHH) and the external validation cohort of 946 men from Parc de Salut Mar (PSM) as well as Hospital Germans Trias i Pujol (GTiPH). This study was conducted between January 1, 2006, and 31 December 2019 [38]. Pre-biopsy 3-Tesla mpMRI and guided or systematic prostate biopsies were always performed. Men undergoing 5-α reductase inhibitor treatment due to symptomatic benign prostatic hyperplasia, having a previous diagnosis of PCa, exhibiting isolated atypical small acinar proliferation, and exhibiting high-grade prostatic intraepithelial neoplasia with atypia were excluded. Furthermore, written consent for prostate biopsy was obtained from all participants, and the project was approved by the institutional review board of VHH (PRAG-317/2017).

### 2.2. Intervention

The development of MRI-PMbdex and the individual generation of csPCa likelihood was expressed as percentages ranging from 0 to 100%. mPSAD (ng/mL^2^) was calculated from the pre-biopsy serum PSA and the prostate volume reported in the pre-biopsy MRI, and individual generation of csPCa likelihood was expressed as percentages ranging from 0 to 100%.

### 2.3. MRI Technique and Evaluation

Magnetic resonance scans were acquired on 3-Tesla scanners with a standard surface phased-array coil. Magnetom Trio (Siemens Corp., Erlangen, Germany) equipment was used in VHH, Diamond Select Achieva (Phillips Corp., Eindoven, The Nederland) in PSM, and Nova Dual (Phillips Corp., Eindoven, Nederland) in GTiPH. The acquisition protocol included T2-weighted imaging (T2W), diffusion-weighted imaging (DWI), and dynamic contrast-enhanced (DCE) imaging, in accordance with the European Society of Urogenital Radiology guidelines [54]. In each institution, an expert radiologist, with over five years of experience and over 300 mpMRI reported per year, analysed the images and reported them according to PI-RADS v2.0, using a 5-point likelihood scale for csPCa [55]. Complex cases were reviewed by two expert radiologists.

### 2.4. Prostate Biopsy Procedure

All men from the participant institutions underwent 2 to 4-core mpMRI-TRUS cognitive fusion-guided biopsies of suspicious lesions and 12-core TRUS systematic biopsies when the PI-RADS reported in the pre-biopsy mpMRI was 3 or higher, while 12-core TRUS systematic biopsies were performed when the PI-RADS was lower than 3 [56]. In each institution, the biopsies were performed by an experienced urologist, with over five years of experience and who had performed over 300 biopsies per year, using a BK Focus 400 ultrasound scanner (BK Medical Inc. Herlev, Denmark) in VHH, Siemens Acuson 150 (Siemens Inc., Erlangen, Germany) in PSM, and a Sonolite Antares (Siemens Inc., Erlangen, Germany) in GTiPH.

### 2.5. Pathologic Analysis and csPCa Definition

The biopsy samples were sent separately to each pathology department, where an expert uro-pathologist analysed the biopsy specimens and, after identifying the PCa, reported the International Society of Uro-Pathology (ISUP) Gleason-based Grade Group (GG) [57]. Complex cases were analysed by two expert uro-pathologists. Any ISUP-GGG ≥ 2 was defined as csPCa.

### 2.6. Endpoint Measurements

The endpoint measurements were the rates of detected csPCa and missed csPCa, in addition to the frequency of avoided prostate biopsies and iPCa.

### 2.7. Statistical Analysis

The reporting recommendations for tumour marker prognostic studies (REMARK) [58] and the update of standards for reporting diagnostic accuracy studies (STARD 2015) [59] were followed. Medians and interquartile ranges (25–75 percentiles) were used to describe the quantitative variables, and rates were used to describe the qualitative variables. The MRI-PMbdex individual likelihoods of csPCa were generated from the logistical regression analysis performed from the included independent predictors of the Barcelona MRI-PM [38]. The mPSAD individual likelihoods of csPCa were generated from the logistic regression analysis performed from the following independent predictors: PI-RADS score and mPSAD. The chi-square and Mann–Whitney tests were used to find associations between the variables. Receiver operating characteristic curves (ROC) and areas under the curve (AUC) were used to analyse the efficacy of mPSAD and MRI-PMbdex for csPCa detection; the DeLong test was conducted to compare the AUCs [60,61]. Furthermore, decision curve analysis (DCA) was used to discern the net benefits of mPSAD and MRI-PMbdex over performing biopsies in all men [62]. The thresholds of mPSAD and MRI-PMbdex were selected from the 95% sensitivity for csPCa, and the specificities were analysed and compared. The performances were analysed based on sensitivity, specificity, negative and positive predictive values, accuracy, avoidable biopsies, and potentially missed csPCa. In addition, odds ratios and 95% confidence intervals (CI) were assessed. Tests with two-sided *p* < 0.05 were considered statistically significant. The statistical analyses were computed using the R programming language v.4.0.3 (The R Foundation for Statistical Computing, Vienna, Austria), and SPSS v.25 (IBM, Statistical Package for Social Sciences, San Francisco, CA, USA) was used.

## 3. Results

### 3.1. Characteristics of the Study Population

The characteristics of the entire population are summarised in Table 1. A median age of 68 years, serum PSA of 6.5 ng/mL, and prostate volume of 55 cm3 can be observed. The DRE was abnormal in 25% of the men; 28% were scheduled to repeat prostate biopsies, and 6.6% had PCa family history. PCa was diagnosed in 1214 men (49.9%), csPCa in 934 men (38.4%), and iPCa in 280 men (11.5%).

The characteristics of the men according to PI-RADS categories as well as comparisons between each pair of consecutive PI-RADS are presented in Table 2. All characteristics showed differences regarding all PI-RADS categories (*p* < 0.001), with the exception of PCa family history, which showed a non-significant trend that ranged from 4.7% in men with PI-RADS 1 to 8.3% in men with PI-RADS 5 (*p* = 0.287). On comparing the characteristics of men with PI-RADS 1 and those with PI-RADS 2, it was found that only the frequency of repeat biopsies was significantly higher among the latter (19.4% vs. 34.1%; *p* < 0.001); all other characteristics were similar between the two categories. Between PI-RADS 2 and PI-RADS 3, the frequency of PCa detection increased from 20.3% to 30.5% (*p* = 0.028), while the frequency of csPCa and iPCa exhibited no significant increase. Men with PI-RADS 4 were significantly older than those with PI-RADS 3, had a higher PSA and mPSAD, had a lower prostate volume, and exhibited an abnormal DRE more often. The frequency of repeat biopsies, PCa family history, and iPCa detection was similar in both PI-RADS 3 and PI-RADS 4 categories; however, a significant increase in csPCa frequency was observed (*p* < 0.001). Moreover, in PI-RADS 5 compared with PI-RADS 4, the frequency of repeat biopsies and iPCa was low, while the frequency of csPCa and overall PCa increased significantly (*p* < 0.001). The frequency of repeat biopsies as well as iPCa also decreased, while the frequency of all PCa and csPCa increased significantly (*p* < 0.001).

### 3.2. Performance of mPSAD and MRI-PMbdex in the Entire Population

The efficacy of MRI-PMbdex in detecting csPCa was higher than that of mPSAD. The AUC was 0.893 (95% CI: 0.880–0.906) for MRI-PMbdex and 0.763 (95% CI: 0.774–0.783) for mPSAD, with *p* < 0.001 (Figure 1A). The MRI-PMbdex exhibited net benefit over mPSAD and over biopsy of all men from a csPCa probability threshold of 2%, while mPSAD exhibited net benefit over biopsy of all men from a csPCa probability threshold of 18% (Figure 1B).

The thresholds of MRI-PMbdex and mPSAD with 95% sensitivity for csPCa were 13.5% and 0.628 ng/mL^2^, respectively, and their corresponding specificities were 51.1% and 19.6%, respectively, with *p* < 0.001 (Table 3). MRI-PMbdex and mPSAD were able to avoid 33.4% and 14.0% of prostate biopsies, respectively, while missing 5% of csPCa (*p* < 0.001). In relation to the aggressiveness of missed csPCa, both mPSAD and MRI-PMbdex missed 2.4% of PCa with ISUP-GG ≥ 3 (22/934). Furthermore, mPSAD missed 1% of csPCa with ISUP-GG ≥ 4 (9/934), while MRI-PMbdex did not miss any of them.

### 3.3. Performance of mPSAD and MRI-PMbdex According to the PI-RADS Categories

Among the 550 men with negative mpMRI (PI-RADS < 3), the MRI-PMbdex exhibited an AUC of 0.824 (95% CI: 0.756–0.892), while mPSAD exhibited an AUC of 0.685 (95% CI: 0.610–0.761), with *p* < 0.001 (Figure 2A). MRI-PMbdex showed net benefit over mPSAD in detecting csPCa (Figure 2B).

The selected thresholds of MRI-PMbdex and mPSAD provided sensitivities of 67.3% and 90.9% as well as specificities of 87.7% and 21.4%, respectively, in men with PI-RADS < 3. The MRI-PMbdex avoided 82.2% of prostate biopsies, whereas mPSAD avoided 20.2%. However, the probabilities of missed csPCa were 32.7% and 9.1%, for MRI-PMbdex and mPSAD, respectively. MRI-PMbdex did not detect 0.9% of PCa with ISUP-GG = 3 (5/55), while mPSAD did not detect 0.2% (1/55) (Table 4).

Among the 645 men with PI-RADS 3, MRI-PMbdex exhibited an AUC of 0.768 (95% CI: 0.717–0.819), while mPSAD exhibited an AUC of 0.696 (95% CI: 0.641–0.750), with *p* < 0.001 (Figure 3A). Thus, MRI-PMbdex showed net benefit over mPSAD from the 18% csPCa probability threshold (Figure 3B).

The selected thresholds of MRI-PMbdex and mPSAD provided sensitivities of 78.9% and 92.7% as well as specificities of 52.1% and 20.5%, respectively, in men with PI-RADS 3. MRI-PMbdex avoided 46.8% of prostate biopsies, while mPSAD avoided 18.3%. However, the probabilities of missed csPCa were 21.7% and 7.3%, respectively. mPSAD did not detect 2.8% of PCa with ISUP-GG ≥ 3 (3/109), whereas MRI-PMbdex did not detect 10% (11/109) (Table 5).

Among the 841 men with PI-RADS 4, the MRI-PMbdex showed an AUC of 0.823 (95% CI: 0.769–0.851), while mPSAD showed an AUC of 0.742 (95% CI: 0.709–0.775), with *p* < 0.001 (Figure 4A). MRI-PMbdex showed net benefit over mPSAD in detecting csPCa from the 20% probability threshold, while mPSAD showed net benefit over performing biopsies on all men from the 35% probability threshold (Figure 4B).

The selected thresholds for MRI-PMbdex and mPSAD provided sensitivities of 98.6% and 94.3% as well as specificities of 12.2% and 16.2%, respectively, in men with PI-RADS 4. MRI-PMbdex avoided 6.5% of prostate biopsies, while mPSAD avoided 10.7%. Moreover, the rate of missed csPCa was 1.4% for MRI-PMbdex and 5.7% for mPSAD. The rate of missed PCa with ISUP-GG ≥ 4 was 0.9% (5/439) for mPSAD and 0% for MRI-PMbdex (Table 6).

Finally, among the 396 men with PI-RADS 5, the MRI-PMbdex showed an AUC of 0.787 (95% CI: 0.728–0.846), while mPSAD showed an AUC of 0.761 (95% CI: 0.698–0.825), with *p* < 0.001 (Figure 5A). Neither MRI-PMbdex nor mPSAD showed net benefit over performing biopsies on all men until a csPCa probability threshold of 65% was reached. In addition, from the 65% probability threshold of csPCa, both MRI-PMbdex and mPSAD showed minimal benefit over performing biopsies on all men (Figure 5B).

The selected thresholds for MRI-PMbdex and mPSAD provided sensitivities of 100% and 97.3% as well as specificities of 4.6% and 20%, respectively, in men with PI-RADS 5. MRI-PMbdex avoided 0.8% of prostate biopsies, while mPSAD avoided 5.6%. The rate of missed csPCa was 0% for MRI-PMbdex and 2.7% for mPSAD. Moreover, mPSAD did not detect 4 PCa with ISUP-GG ≥ 4 (Table 7).

## 4. Discussion

First, the characteristics of suspected PCa regarding PI-RADS categories are considered. The similar characteristics of men with PI-RADS 1 and PI-RADS 2, in addition to similar csPCa detection rates (9.8% vs. 10.6%) and high iPCa over detection rates (above 50%) in both subsets, justify considering these two PI-RADS categories as a negative mpMRI. The observed 90% negative predictive value of mpMRI was in the recently reported ranges (i.e., 80–95%) [10,11,12]. The characteristics of suspected PCa with PI-RADS 3 were closer to those with PI-RADS 2 than those with PI-RADS 4. Moreover, the csPCa detection rate with PI-RADS 3 (16.9%) was closer to that of men with PI-RADS 2 (10.8%) than that of men with PI-RADS 4 (52.2%). Therefore, PI-RADS 3 is an uncertain scenario more similar to a negative MRI than to PI-RADS 4; this justifies why some authors consider men with PI-RADS < 4 as candidates to avoid prostate biopsy [9,15]. Finally, the characteristics of those with PI-RADS 5 were notably different from those with PI-RADS 4. Furthermore, the csPCa risk increased with PI-RAD 5, reaching 83.6%. Taking these factors into account, it is clear that considering PI-RADS > 3 in the same group as the other categories is inadvisable. Hence, the data of men with PI-RADS 4 and PI-RADS 5 should be considered separately [5].

PSAD was introduced by Benson et al. in 1992 to improve the specificity of serum PSA in distinguishing men with localised PCa from those with benign prostatic enlargement [17,63]. Shortly after, it was incorporated by Catalona et al. as a tool for prostate biopsy decision-making in the early detection of PCa [64]. The inverse relationship between prostate volume and PCa risk has been recently confirmed in a systematic review of the literature from the last thirty years [65]. Transrectal ultrasonography was recommended for assessing the prostate volumetry for PSAD calculation from the beginning, as it yields a more accurate assessment than suprapubic ultrasonography [66]. Most studies analysing the clinical usefulness of PSAD have assessed the prostate volume just before TRUS systematic biopsies [67,68,69,70]. However, routine TRUS prostate volume assessment for calculating PSAD is not typically performed for prostate biopsy decision-making. The spread ERSPC (European Randomised Screening Prostate Cancer) risk calculator has incorporated the highly subjective prostate volume estimation from DRE [71,72]. The main reason for the current resurgence of PSAD may be the spread of pre-biopsy MRI [73], which provides the most accurate prostate volume assessment without additional cost; this helps avoid TRUS, which is time-consuming and cumbersome [74]. mPSAD has been directly incorporated as ng/mL^2^ into some MRI-PM [23,26,32,33,75] or indirectly incorporated from serum PSA and prostate volume into other MRI-PM [28,38,76,77,78,79]. Logistical regressions performed to develop MRI-PM have shown PSAD to be the most powerful predictor of csPCa after the PI-RADS score. Recent studies analysing the clinical usefulness of mPSAD for csPCa detection according to PI-RADS categories have shown its dynamic behaviour, which suggests the need for different thresholds to obtain similar predictive values in different PI-RADS categories [21,80]. Nevertheless, the proper mPSAD thresholds for specific PI-RADS categories should be selected from each area, given the specific characteristics of cared populations of men with suspected PCa [22,81]. The most recent MRI-PMs share the latest PI-RADS versions, with direct or indirect mPSAD, and may share other independent clinical predictors, such as age, PCa family history, initial or repeat biopsy, and DRE.

Our study is the first to compare mPSAD and MRI-PM in a head-to-head manner for improving the selection of candidates for prostate biopsy in suspected PCa men after mpMRI. The compared MRI-PMbdex shared the same predictors of csPCa as the recent Barcelona MRI-PM. MRI-PMbdex and mPSAD avoid the inconveniences of modern markers as they are free of cost, do not require sample procurement, and require very little time for the assessment. This study was carried out on a sizable, multicentre population representing a metropolitan area. In all participant institutions, the criteria for PCa suspicion and diagnostic approach adhered to the EAU PCa guidelines [7]. MRI-PMbdex outperformed mPSAD in the entire population, and it exhibited net benefit over mPSAD, which, in turn, exhibited a slight benefit over performing biopsies on all men. The behaviour of both tools regarding PI-RADS categories showed that MRI-PMbdex outperformed mPSAD in each PI-RADS ≤ 4 category, with different degrees of benefit. However, clinical benefit was found for any tool in men with PI-RADS 5. Additionally, mPSAD did not detect more aggressive tumours compared to MRI-PMbdex in men with PI-RADS ≥ 4. The different behaviour of mPSAD [21] and MRI-PMs [38] regarding PI-RADS categories is the consequence of the incidence of csPCa among them. We selected different thresholds of mPSAD to obtain similar performance in our series of PI-RADS 3 than that of Görtz et al. [22,81]. The use of appropriate thresholds in different populations has been previously described for mPSAD [19,20,21,82,83,84,85,86,87,88,89,90,91,92,93,94,95].

The limitations of this study include the design, which is prone to sampling and selection biases. Furthermore, no concordance analysis was performed by the pathologists, radiologists, and urologists involved in the study. In addition, the inter-variability that is inevitable among researchers involved in multicentric studies also exists. Moreover, prostate biopsies and mPSAD suffer from intrinsic limitations related to the heterogeneity of prostate cancer and prostate volume. The lack of histopathological analysis of surgical specimens is another limitation, although not every PCa patient undergoes radical prostatectomy. While the definition we adopted for csPCa is the most common one, it may not represent the true csPCa. Moreover, as the MRI-PMbdex is not the developed Barcelona MRI-PM, the results cannot be attributed to this model. Hence, further studies analysing the clinical usefulness of mPSAD and MRI-PMs according to PI-RADS categories are needed.

It will be important in the future to include the results of radiomics and radio-genomics to improve the next generation of MRI-PMs. It appears likely that we will have more powerful predictors than PI-RADS and a more refined definition of csPCa based on genomic expression in addition to the current the pathology of prostate biopsies or surgical specimens [96,97].

## 5. Conclusions

MRI-PMbdex, which used the same predictors as the recent Barcelona MRI-PM, outperformed mPSAD in the proper selection of candidates for prostate biopsy. MRI-PMbdex exhibited net benefit over mPSAD in men with PI-RADS ≤ 4. However, regarding men with PI-RADS 5, neither tool showed clinical benefit over biopsy. The clinical usefulness of tools that improve the selection of candidates for prostate biopsy in the entire population does not represent those observed in each PI-RADS category.

## Figures and Tables

**Figure 1 cancers-14-02374-f001:**
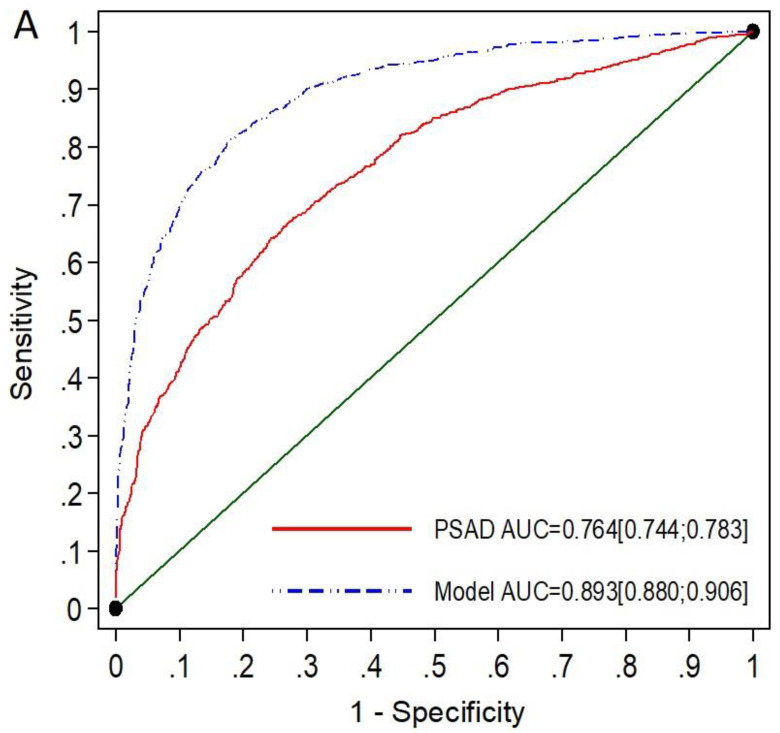
The ROC curves and AUCs show the efficacy of mPSAD (PSAD in the graphic) and MRI-PMbdex (model in the graphic) in detecting csPCa (**A**). The DCAs show the net benefit of mPSAD and MRI-PMbdex compared to performing biopsies on all men according to the csPCa probability threshold (**B**).

**Figure 2 cancers-14-02374-f002:**
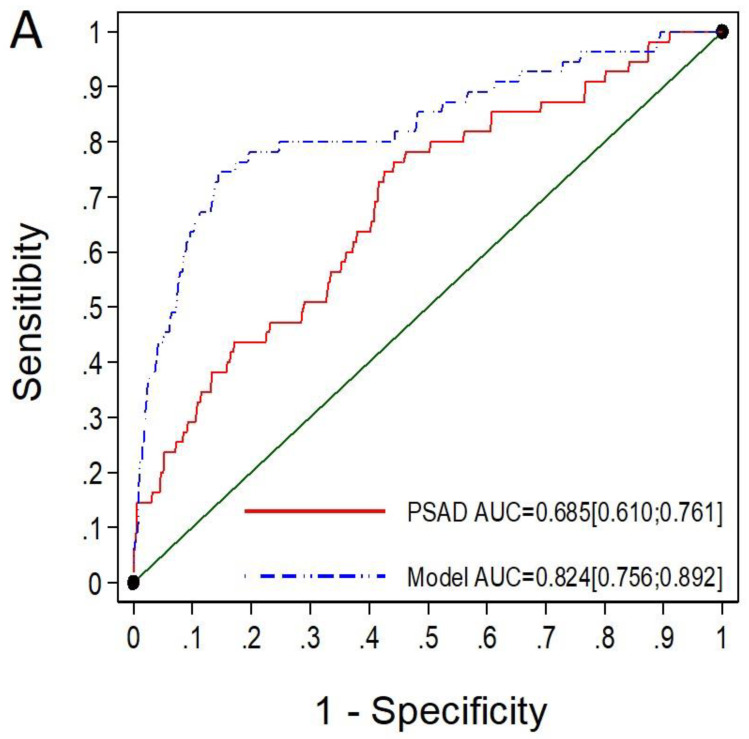
The ROC curves and AUCs show the efficacy of mPSAD (PSAD in the graphic) and MRI-PMbdex (model in the graphic) in detecting csPCa in men with PI-RADS < 3 (**A**). The DCAs showing the net benefit of mPSAD and MRI-PMbdex over performing biopsies on all men according to the csPCa probability threshold (**B**).

**Figure 3 cancers-14-02374-f003:**
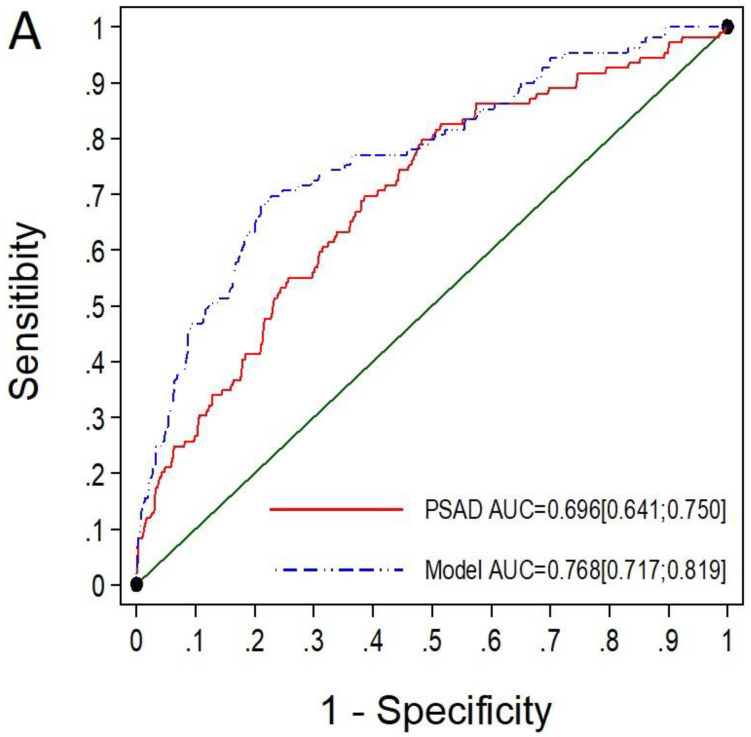
The ROC curves and AUCs show the efficacy of mPSAD (PSAD in the graphic) and MRI-PMbdex (model in the graphic) in detecting csPCa in men with PI-RADS 3 (**A**). The DCAs showing the net benefit of mPSAD and MRI-PMbdex over performing biopsies on all men according to the csPCa probability threshold (**B**).

**Figure 4 cancers-14-02374-f004:**
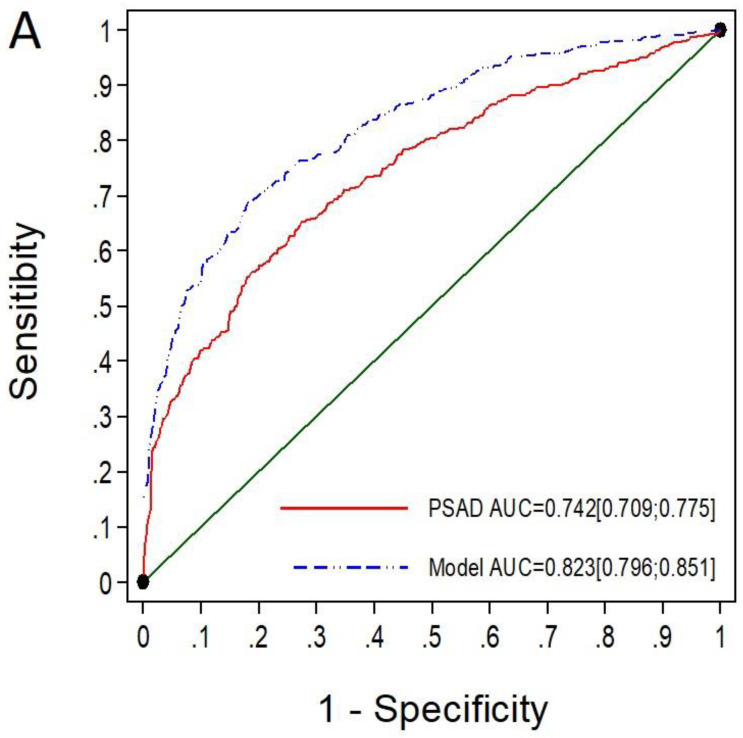
The ROC curves and AUCs show the efficacy of mPSAD (PSAD in the graphic) and MRI-PMbdex (model in the graphic) in detecting csPCa in men with PI-RADS 4 (**A**). The DCAs showing the net benefit of mPSAD and MRI-PMbdex over performing biopsies on all men according to the csPCa probability threshold (**B**).

**Figure 5 cancers-14-02374-f005:**
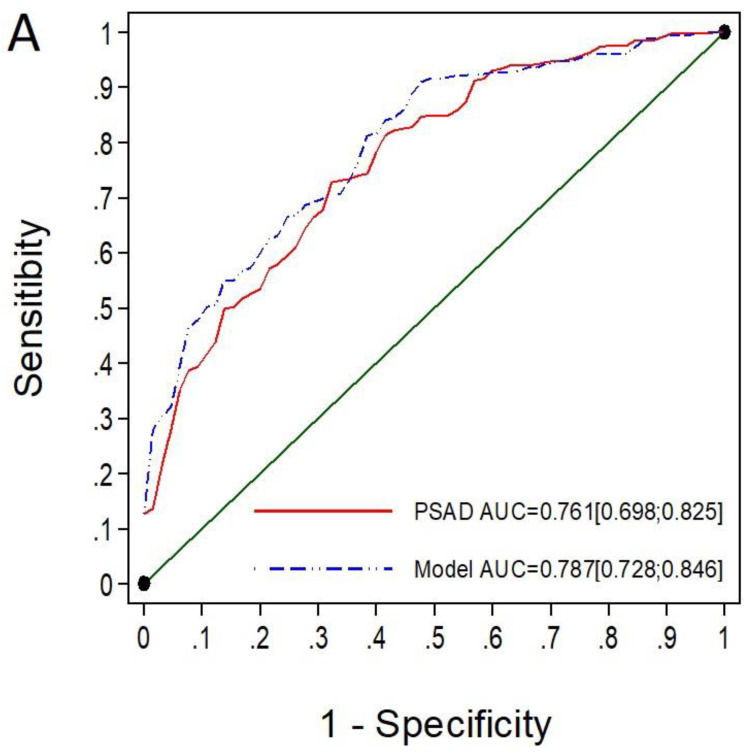
The ROC curves and AUCs show the efficacy of mPSAD (PSAD in the graphic) and MRI-PMbdex (model in the graphic) in detecting csPCa in men with PI-RADS 5 (**A**). The DCAs showing the net benefit of mPSAD and MRI-PMbdex over performing biopsies on all men according to the csPCa probability threshold (**B**).

**Table 1 cancers-14-02374-t001:** Characteristics of study population.

Characteristic	Measurement
Number of cases	2432
Median age, years (IQR)	68 (62–73)
Median total PSA, ng/mL (IQR)	6.5 (4.7–10.0)
Abnormal DRE, *n* (%)	612 (25.2)
Median prostate volume, ml (IQR)	55 (40–77)
Median PSA density, ng/mL^2^ (IQR)	0.12 (0.08–0.20)
Repeat biopsy, *n* (%)	681 (28.0)
Family history of PCa, *n* (%)	161 (6.6%)
PI-RADS, *n* (%)	
1–2	550 (22.6)
3	645 (26.5)
4	841 (34.6)
5	396 (16.3)
Overall PCa detection, *n* (%)	1214 (49.9)
csPCa detection, *n* (%)	934 (38.4)
iPCa detection, *n* (%)	280 (11.5)

**Table 2 cancers-14-02374-t002:** Characteristics of study population according to the PI-RADS category.

Characteristic	PI-RADS 1	*p* Value	PI-RADS 2	*p* Value	PI-RADS 3	*p* Value	PI-RADS 4	*p* Value	PI-RADS 5
Number of cases	427	-	123	-	645	-	841	-	396
Median age, years (IQR)	66 (60–72)	=0.633	66 (60–71)	=0.901	66 (60–71)	<0.001	69 (63–74)	<0.001	72 (66–76)
Median total PSA, ng/mL (IQR)	6.2 (4.5–8.7)	=0.323	5.8 (4.4–8.4)	=0.644	6.0 (4.4–8.4)	<0.001	6.6 (4.8–9.5)	<0.001	9.4 (5.8–18.0)
Abnormal DRE, *n* (%)	59 (13.8)	=0.818	18 (14.6)	=0.497	80 (12.4)	<0.001	216 (25.7)	<0.001	239 (60.4)
Median PV, mL (IQR)	63 (43–90)	=0.666	60 (45–84)	=0.833	63 (45–82)	<0.001	50 (37–74)	<0.001	46 (35–60)
Median PSAD, ng/mL^2^ (IQR)	0.10 (0.07–0.15)	=0.960	0.11 (0.07–0.15)	=0.797	0.10 (0.07–0.15)	<0.001	0.13 (0.08–0.20)	<0.001	0.21 (0.13–0.37)
Repeat biopsy, *n* (%)	83 (19.4)	<0.001	42 (34.1)	=0.631	206 (31.9)	=0.860	271 (32.3)	<0.001	78 (19.7)
Family history of PCa, *n* (%)	20 (4.7)	=0.649	7 (5.7)	=0.733	42 (6.5)	=0.702	59 (7.0)	=0.410	33 (8.3)
Overall PCa detection, *n* (%)	97 (22.7)	=0.574	25 (20.3)	=0.028	194 (30.5)	<0.001	548 (65.2)	<0.001	350 (88.4)
csPCa detection, *n* (%)	42 (9.8)	=0.865	13 (10.6)	=0.081	109 (16.9)	<0.001	439 (52.2)	<0.001	331 (83.6)
iPCa detection, *n* (%)	55 (12.9)	=0.351	12 (9.8)	=0.295	85 (13.2)	=0.902	109 (13.0)	<0.001	19 (4.8)

**Table 3 cancers-14-02374-t003:** Performance of mPSAD and MRI-PMbdex, with respective thresholds of 0.628 ng/mL^2^ and 13.5%, in the entire study population and grade groups of detected tumours.

Parameter	mPSAD	MRI-PMbdex
Sensitivity	887/934 (95.0)	887/934 (95.0)
Specificity	294/2091 (19.6)	765/1498 (51.1)
Negative predictive value	294/341 (86.2)	765/812 (94.2)
Positive predictive value	887/2091 (42.4)	887/1620 (54.8)
Accuracy	1181/2432 (48.6)	1652/2432 (67.9)
Avoided biopsies	341/2432 (14.0)	812/2432 (33.4)
Missed csPCa	47/934 (5%)	47/934 (5%)
Odds ratios (95% CI)	4.61 (3.35–6.35)	19.70 (14.44–26.86)
*p* Value	<0.001	<0.001
GG 2	25	25
GG 3	13	22
GG 4	8	0
GG 5	1	0

**Table 4 cancers-14-02374-t004:** Performance of mPSAD and MRI-PMbdex, with respective thresholds of 0.628 ng/mL^2^ and 13.5%, in men with PI-RADS < 3.

Parameter	mPSAD	MRI-PMbdex
Sensitivity	50/55 (90.9)	37/55 (67.3)
Specificity	106/495 (21.4)	434/495 (87.7)
Negative predictive value	106/111 (95.5)	434/452 (96.0)
Positive predictive value	50/439 (11.4)	37/98 (67.3)
Accuracy	156/550 (28.4)	471/550 (85.6)
Avoided biopsies	111/550 (20.2)	452/550 (82.2)
Missed csPCa	5/55 (9.1)	18/55 (32.7)
Odds ratio (95% CI)	2.27 (1.06–7.00)	14.62 (7.84–27, 29)
*p* Value	=0.033	<0.001
GG 2	4	13
GG 3	1	5
GG 4	0	0
GG 5	0	0

**Table 5 cancers-14-02374-t005:** Performance of mPSAD and MRI-PMbdex, with respective thresholds of 0.628 ng/mL^2^ and 13.5%, in men with PI-RADS 3.

Parameter	mPSAD	MRI-PMbdex
Sensitivity	101/109 (92.7)	86/109 (78.9)
Specificity	110/536 (20.5)	279/536 (52.1)
Negative predictive value	110/118 (93.2)	279/302 (92.4)
Positive predictive value	101/527 (19.2)	86/343 (25.1)
Accuracy	211/645 (32.5)	365/645 (56.6)
Avoided biopsies	118/645 (18.3)	302/645 (46.8)
Missed csPCa	8/109 (7.3)	23/109 (21.1)
Odds Ratio (95% CI)	3.26 (1.54–6.90)	4.06 (2.49–6.63)
*p* Value	<0.001	<0.001
GG 2	5	12
GG 3	2	11
GG 4	1	0
GG 5	0	0

**Table 6 cancers-14-02374-t006:** Performance of mPSAD and MRI-PMbdex, with respective thresholds of 0.628 ng/mL^2^ and 13.5%, in men with PI-RADS 4.

Parameter	mPSAD	MRI-PMbdex
Sensitivity	414/439 (94.3)	433/439 (98.6)
Specificity	65/402 (16.2)	42/402 (12.2)
Negative predictive value	65/90 (72.2)	49/55 (89.1)
Positive predictive value	414/751 (55.1)	433/786 (55.1)
Accuracy	479/841 (57.0)	482/841 (57.3)
Avoided biopsies	90/841 (10.7)	55/841 (6.5)
Missed csPCa	25/439 (5.7)	6/439 (1.4)
Odds Ratio (95% CI)	3.19 (1.97–5.18)	10.02 (4.24–23.66)
*p* Value	<0.001	<0.001
GG 2	12	1
GG 3	8	5
GG 4	5	0
GG 5	0	0

**Table 7 cancers-14-02374-t007:** Performance of MRI-PSAD and Barcelona MRI-predictive model in men with PI-RADS 5, from the respective thresholds of 0.628 ng/mL^2^ and 13.5%.

Parameter	mPSAD	MRI-PMbdex
Sensitivity	322/331 (97.3)	331/331 (100)
Specificity	13/65 (20.0)	3/65 (4.6)
Negative predictive value	13/22 (59.1)	3/3 (100)
Positive predictive value	322/374 (86.1)	331/393 (84.2)
Accuracy	335/396 (84.6)	334/396 (84.3)
Avoided biopsies	22/396 (5.6)	3/396 (0.8)
Missed csPCa	9/331 (2.7)	0/331 (0)
Odds Ratio (95% CI)	8.94 (3.63–21.98)	-
*p* Value	<0.001	=0.004
GG 2	2	0
GG 3	3	0
GG 4	3	0
GG 5	1	0

## Data Availability

The data presented in this study are available on request from the corresponding author.

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
