# Peer review of "Comparative Analysis of PSA Density and an MRI-Based Predictive Model to Improve the Selection of Candidates for Prostate Biopsy"

_cancers, 2022, doi:10.3390/cancers14102374_

Round 1
Reviewer 1 Report
This paper has important content but the reading for me was painful. I would consider a manuscript in this state to be a draft. Although I have offered corrections into the results section, I have had to stop due to the frequency of changes I believe are necessary and the amount of time that will be required on my part to complete my review. In forty years of reviewing articles for JAMA, J Urol, Urology, J Clin Onc, I have never "quit" in the midst of a review. The lead author has written many outstanding papers and I cannot recall any that were as onerous to read as this presented manuscript. There is definitely a need for formal input from a medically trained person who is excellent in using the English language. I do not expect any credit for my partial review since I did not complete the mission. I apologize.
Review of manuscript by Juan Morote et al: Head-to-Head Comparison of Magnetic Resonance Imaging-Derived Prostate-Specific Antigen Density and a Predictive
Model Based on Magnetic Resonance Imaging for the Proper Selection of Candidates for Prostate Biopsy in Men with Suspected Prostate Cancer.
First, the overall conclusion of this paper is significant, and the paper is quite worthy of being published. With that said, I will say that the sentence structure and overall grammar is extremely poor, and this makes this paper burdensome to read. Articles are missing, prepositions are incorrectly used and dangling modifiers occur. The authors need to address this point first and foremost and consider involving someone with a far better command of the written English language.
Second, providing a list of abbreviations used in the manuscript at the front end of the publication would be helpful to the readers, as well as ensuring the authors do not repeat expanding the abbreviation throughout the text as in the original manuscript. The convention is that once an abbrevation such as digital rectal examination (DRE) is used, then simply use DRE in any subsequent comments involving this phrase. Also, if there are earlier papers that have established an abbrevation, it helps to use that since the reader is already familiar with the abbreviation. For years, PSA density (PSAD) has been used. Now, I see some authors altering this to PSAd. Let’s make these conventions consistent. A major paper has used mPSAD for PSA density obtained using the mp-MRI-based volume. That abbreviation is a lot simpler than MRI-PSAD.
My critique will use the line numbers on the manuscript.
4-6: the title is wordy & readability can be increased by shortening:
Head-to-Head Comparison of Magnetic Resonance Imaging-Derived Prostate-Specific Antigen Density and an Adjusted Predictive Model for Selecting Biopsy Candidates in Men with Clinically Suspected Prostate Cancer.
Simple Summary:
SS: recommend authors include section with abbreviations and that they are consistent with their use of the abbreviations throughout the publication.
30: suggest use an already established abbreviation (e.g., mPSAD) for MRI-associated PSAD. Of course, only the volume determination in the PSAD is from the mp-MRI and a more correct phrase is MRI-volume PSAD and not “derived”.
31: In other papers involving very large patient numbers, the MRI-based PSAD is abbreviated mPSAD. It would be great if we could all agree on the same abbreviations (i.e., Niu, your reference 23, calls it adjusted-PSAD to define MRI-based PV-adjusted Prostate Specific Antigen Density (adjusted-PSAD)).
33: with suspected prostate cancer (PCa). There is growing confusion in the abbreviations relating to significant prostate cancer. I would not show (PCa) after referring to suspected prostate cancer on line 33. Show that abbreviation later or assume everyone knows that PCa equates with prostate cancer. The abbreviations csPCa and sPCa confuse the reader. If one is chosen, define it. Some papers are using Gleason Grade Group (GGG) 2 or higher to mean csPCa. Others are using GGG 3 or higher (one of the references you provided). Others use GGG 1 with a specific core length showing prostate cancer. I personally do not think it is necessary to explain definitions such as DRE, PSA, TRUS or MRI.
36: grammar: change (consider) to: We aim to compare the clinical usefulness of mPSAD and a new MRI predictive model that utilizes the same predictors as the recently developed and externally validated Barcelona MRI predictive model (MRI-PMbdex).
SS: this way, the reader is always clear as to what you are referring to. And throughout the paper, when you refer to the adjusted MRI-PM use MRI-PMbdex. The bdex refers to Barcelona development and external validation.
38: If I understand this manuscript correctly, the following sentence should be accurate: This study is a head-to-head comparison between mPSAD vs. an adjusted MRI-PM that was created from a study of 2,432 men with suspected prostate cancer. This cohort resulted in the creation of a predictive model that underwent development and external validation and was named the Barcelona MRI-PM, and now abbreviated as (MRI-PMb).
40-42: This is a very bizarre “sentence.” What are you trying to say here? You state that pre-biopsy mp-MRI and TRUS-guided biopsies “to Prostate Imaging Reporting and Data System (PI-RADS) ≥ 3 lesions and/or 12-core TRUS systematic biopsy were scheduled.” What do you mean by “to”? Please clarify with a rewrite.
43-46: see changes on submitted manuscript
46: clarify if the 18% is greater than or greater than and equal to
48: are you sure that with a mPSAD of 0.628 the specificity was only 19.6%? That seems very low for this PSAD.
49: “while any of them” please clarify: do you mean either mPSAD or MRI-PMbdex (both)? See possible alternative sentences in the annotated manuscript.
50-52: sentence “We conclude…” is should be corrected to read We conclude that the MRI-PMbdex was more accurate than mPSAD for the proper selection of candidates for prostate biopsy in men with suspected PCa, with an exception in men with a PI-RADS 5 score in whom neither tool exhibited clinical guidance to determine the need for biopsy. …..
57-59: see the annotated manuscript for one possible sentence to replace the original text.
68-72: see annotated manuscript. Also, besides changes shown consider: PI-RADS 3 justifies its description as “equivocal” because in this context the detection of csPCa usually does not reach 20% and because more than 50% of biopsied lesions represent iPCa [9,15].
76-78: alter sentence to: However, PSAD has now been strengthened by pre-biopsy MRI, which provides a more accurate prostate volume at no additional cost [18].
78-79: This is not as accurate a statement as it should be. Suggest: mPSAD has been analyzed in overall populations of men to enhance the detection of csPCa in relation to the PI-RADS score [19-22]. [I do not even think the word “overall” is needed or makes sense.
80-83: this is too wordy. Shorten to: MRI-PMs are also attractive tools. However, few MRI-PMs have been developed from the latest versions of PI-RADS and external validation should be performed before their use. Easily accessible risk calculators (RC) are essential to avoid the cumbersome and often time-consuming nomograms [23-27].
Note: I do not see any nomogram presented in paper by Mehralivand (ref. 26). I agree that the nomogram in the Truong paper (ref. 27) is onerous. I do not find the Lee nomogram very time-consuming or cumbersome. Also, all nomograms are able to be automated or converted into a computer calculator and made accessible on the internet. You may wish to note this in your discussion. An example of one such program is at http://www.prostatecalculator.org although this calculator does not relate to mp-MRI or mPSAD. Another excellent site is run by Tom Hueting of The Netherlands. See www.evidencio.com.
83: suggest distinguish the original Barcelona MRI-PM with the designation MRI-PMb (see annotation in margin for section 2.1 in the manuscript). I cannot tell from line 83 when you state “recent Barcelona MRI-PM if you mean the one already published or the one that is the subject of this paper. It appears that it is the original and already published one (ref. 38 ⇢ Morote J, Borque-Fernando A, Triquell M, Celma A, Regis L, Abascal JM, Sola C, Servian P, Escobar M, Mast R, et al. The Barcelona Predictive Model of Clinically Significant Prostate Cancer. Cancers. 2022, 14, 1589). It appears you do reference this on line 90. Perhaps moving that citation to line 86 would make it clearer to the reader which MRI-PM you are referring to.
92: clarify meaning of “and outperformance of the model in specific PI-RADS categories. Do you mean: and in the future to improve the performance of the model in specific PI-RADS categories.?
96: Instead of “been analyzed after mp-MRI showing… consider “have improved the prognostic performance of mpMRI.”
98: Once again, a clearer statement is needed here.
The aim of this study is to compare the clinical usefulness of mPSAD with MRI-PMbdex for the proper….
Section 2.1 (103-107): see annotations on manuscript.
104: incorrect English. Lines 103-105 would be better as: A head-to-head comparison of mPSAD vs. an adjusted MRI-PM (MRI-PMbdex) was designed from a multicenter trial conducted in the metropolitan area of Barcelona. The trial involved 2,432 men with suspected prostate cancer due to a serum PSA >3 ng/ml and/or an abnormal DRE.
105: The adjusted MRI-PM (MRI-PMbdex) was created from the development cohort of 1,486 men studied at the Vall d’Hebron Hospital (VHH) and the external validation cohort of 946 men from Parc de Salut Mar (PSM) and the Hospital Germans Trias I Pujol (GTiPH). That study was conducted between January 1 of 2006 and December 31 of 2019 [38].
129: no need for commas after radiologists or after institution.
134-136: confusing as written; please clarify.
- Results
176-179: The DRE was abnormal in 25% of men; 28% had a prior negative biopsy and were scheduled for repeat biopsies, and 6.6% had a PCa family history. PCa was diagnosed in 1,214 men (49.9%), csPCa in 934 (38.4%) and iPCa in 280 (11.5%).
Table 1. median PSAD is shown as ng/ml/cc. Density is weight per unit volume. The units should be ng/ml or ng/cc which is the same with PSAD. It should not be ng/ml/cc.
184: comma should be after the word categories.
186: change to: …except PCa family history, which showed a non-significant trend that was 4.7% with PI-RADS 1 and 8.3% for PI-RADS 5 (p =0.31).
Lines 189, 191-2: I believe you are looking at “frequency” rather than rate. : rate is defined as a quantity, amount, or degree of something measured per unit of something else. Other minor suggested changes shown.
193: suggest “Men with PI-RADS 4 were of a significantly higher age than those with PI-RADS 3, had a higher PSA and PSAD, a lower prostate volume, and more often had an abnormal DRE.”
195-197: again, it is frequencies rather than rates. Also clarify what PI-RADS groups you are referring to by changing (line 195) “were similar in PI-RADS 3 and 4 as was the frequency of iPCa. In the next sentence I do not know which PI-RADS category (3 or 4) had the higher frequency of csPCa. You need to clarify that. Other changes shown in annotations.
Line 199: suggest change to: Also, in the PI-RADS 5 category, the frequency of repeat biopsy was less as was that of iPCa, while the frequency of csPCa and overall PCa increased significantly (P <0.001).
208-212: see annotations in manuscript
Figures 1A and 1B: I think the legend is too small insofar as font size and will be difficult to read in the published article. Also, the legend for 1A shows PSAD and it is the mPSAD or MRI-PSAD, while the Model should be labeled as MRI-PMbdex or whatever you decide is the right abbrevation. The same is the case for the font size of the legend in 1B as well as the legend description. I honestly do not understand the item “Net Benefit Treat All”. The legend for the figure and the text in the manuscript refers to “biopsy all men”. If so, there should be consistency between text, legend and figure caption. In the legend, you do not expand the definition “DCA”.
I have spent far too many hours trying to understand what the writer(s) have said with a significant part of this time amending sentence structure and grammar. I did show some additional editing as annotations in the manuscript.
I am also attaching the authors’ manuscript with additional comments shown as annotations.

Author Response
Dear Reviewer, I feel embarrassed because my English is so bad. This should not be a justification for presenting a manuscript so poorly. I apologise to this reviewer and the editor of Cancers for this lack of respect for the rules.
I have commissioned a thorough revision of the English by a specialised company, and I hope that the manuscript has improved sufficiently.
I appreciate your comments and criticisms of the study very much. I have made all the changes that you suggest in your review.
Respectfully,
Juan Morote
Review of manuscript by Juan Morote et al: Head-to-Head Comparison of Magnetic Resonance Imaging-Derived Prostate-Specific Antigen Density and a Predictive Model Based on Magnetic Resonance Imaging for the Proper Selection of Candidates for Prostate Biopsy in Men with Suspected Prostate Cancer.
First, the overall conclusion of this paper is significant, and the paper is quite worthy of being published. With that said, I will say that the sentence structure and overall grammar is extremely poor, and this makes this paper burdensome to read. Articles are missing, prepositions are incorrectly used and dangling modifiers occur. The authors need to address this point first and foremost and consider involving someone with a far better command of the written English language.
I am sorry for this inconvenience. An expert English Proofreading has been preformed
Second, providing a list of abbreviations used in the manuscript at the front end of the publication would be helpful to the readers, as well as ensuring the authors do not repeat expanding the abbreviation throughout the text as in the original manuscript. The convention is that once an abbreviation such as digital rectal examination (DRE) is used, then simply use DRE in any subsequent comments involving this phrase. Also, if there are earlier papers that have established an abbreviation, it helps to use that since the reader is already familiar with the abbreviation. For years, PSA density (PSAD) has been used. Now, I see some authors altering this to PSAd. Let’s make these conventions consistent. A major paper has used mPSAD for PSA density obtained using the mp-MRI-based volume. That abbreviation is a lot simpler than MRI-PSAD.
Thanks. The list of acronyms has been created. In addition, your suggestion of using mPSAD has been incorporated in the list, and the manuscript.
Following this suggestion, I have also changed the acronym of MRI-based predictive model by MRI-PMbdex.
My critique will use the line numbers on the manuscript.
4-6: the title is wordy & readability can be increased by shortening:
Thanks, I agree and propose: “Comparative analysis of PSA Density and an MRI-based Predictive Model to improve the Selection of Candidates for prostate Biopsy”
Simple Summary:
SS: recommend authors include section with abbreviations and that they are consistent with their use of the abbreviations throughout the publication.
Thanks. It has been included
30: suggest use an already established abbreviation (e.g., mPSAD) for MRI-associated PSAD. Of course, only the volume determination in the PSAD is from the mp-MRI and a more correct phrase is MRI-volume PSAD and not “derived”.
OK, No line 29-30 is “Magnetic resonance imaging-associated prostate-specific antigen density (mPSAD)”
31: In other papers involving very large patient numbers, the MRI-based PSAD is abbreviated mPSAD. It would be great if we could all agree on the same abbreviations (i.e., Niu, your reference 23, calls it adjusted-PSAD to define MRI-based PV-adjusted Prostate Specific Antigen Density (adjusted-PSAD)).
OK, now mPSAD is adapted as acronym in all the manuscript
33: with suspected prostate cancer (PCa). There is growing confusion in the abbreviations relating to significant prostate cancer. I would not show (PCa) after referring to suspected prostate cancer on line 33. Show that abbreviation later or assume everyone knows that PCa equates with prostate cancer. The abbreviations csPCa and sPCa confuse the reader. If one is chosen, define it. Some papers are using Gleason Grade Group (GGG) 2 or higher to mean csPCa. Others are using GGG 3 or higher (one of the references you provided). Others use GGG 1 with a specific core length showing prostate cancer. I personally do not think it is necessary to explain definitions such as DRE, PSA, TRUS or MRI.
Thank you, abbreviations are corrected as you suggest
Line 39 is now: “Clinically significant PCa (csPCa), defined when Gleason Grade Group 2 or higher”
36: grammar: change (consider) to: We aim to compare the clinical usefulness of mPSAD and a new MRI predictive model that utilises the same predictors as the recently developed and externally validated Barcelona MRI predictive model (MRI-PMbdex).
OK, the sentence is changed: We aim to compare the clinical usefulness of mPSAD and a new MRI predictive model that utilises the same predictors as the recently developed and externally validated Barcelona MRI predictive model (MRI-PMbdex).
SS: this way, the reader is always clear as to what you are referring to. And throughout the paper, when you refer to the adjusted MRI-PM use MRI-PMbdex. The bdex refers to Barcelona development and external validation.
Thank you very much for the suggestion. Changes have been done
38: If I understand this manuscript correctly, the following sentence should be accurate: This study is a head-to-head comparison between mPSAD vs. an adjusted MRI-PM that was created from a study of 2,432 men with suspected prostate cancer. This cohort resulted in the creation of a predictive model that underwent development and external validation and was named the Barcelona MRI-PM, and now abbreviated as (MRI-PMb).
Yes, your appreciation is correct. The sentence is now “This study is a head-to-head comparison between mPSAD vs. an adjusted MRI predictive model that was created from a study of 2,432 men with suspected prostate cancer. This cohort resulted in the development and external validation cohorts of MRI-PMbdex).
40-42: This is a very bizarre “sentence.” What are you trying to say here? You state that pre-biopsy mp-MRI and TRUS-guided biopsies “to Prostate Imaging Reporting and Data System (PI-RADS) ≥ 3 lesions and/or 12-core TRUS systematic biopsy were scheduled.” What do you mean by “to”? Please clarify with a rewrite.
OK, the sentence has been rewritten “Pre-biopsy 3-Tesla multiparametric MRI (mpMRI) and 2 to 4-core transrectal ultrasound (TRUS) guided biopsies to suspicious lesions and/or 12-core TRUS systematic biopsy were scheduled.”
43-46: see changes on submitted manuscript
Changes on submitted manuscript are in red color
46: clarify if the 18% is greater than or greater than and equal to
Sentence is changed: “MRI-PMbdex showed net benefit over biopsy all men when the probability of csPCa was greater than 2%, while mPSAD did so when the probability probability of csPCa was greater than 18%”
48: are you sure that with a mPSAD of 0.628 the specificity was only 19.6%? That seems very low for this PSAD.
Yes, as it is indicated in table 3: 294/2091 (19.6)
49: “while any of them” please clarify: do you mean either mPSAD or MRI-PMbdex (both)? See possible alternative sentences in the annotated manuscript.
The sentence has been rewrite: “MRI-PMbdex net benefit over mPSAD in men with PI-RADS <4, while none exhibited benefit in men with PI-RADS 5.”
50-52: sentence “We conclude…” is should be corrected to read We conclude that the MRI-PMbdex was more accurate than mPSAD for the proper selection of candidates for prostate biopsy in men with suspected PCa, with an exception in men with a PI-RADS 5 score in whom neither tool exhibited clinical guidance to determine the need for biopsy. …..
This is the new sentence: “We conclude that the MRI-PMbdex was more accurate than mPSAD for the proper selection of candidates for prostate biopsy in men with suspected PCa, with an exception in men with a PI-RADS 5 score in whom neither tool exhibited clinical guidance to determine the need for biopsy.”
57-59: see the annotated manuscript for one possible sentence to replace the original text.
OK
68-72: see annotated manuscript. Also, besides changes shown consider: PI-RADS 3 justifies its description as “equivocal” because in this context the detection of csPCa usually does not reach 20% and because more than 50% of biopsied lesions represent iPCa [9,15].
OK
76-78: alter sentence to: However, PSAD has now been strengthened by pre-biopsy MRI, which provides a more accurate prostate volume at no additional cost [18].
Current lines 73-74 changed: “However, PSAD has now been strengthened by pre-biopsy MRI, which provides a more accurate prostate volume at no additional cost [18].”
78-79: This is not as accurate a statement as it should be. Suggest: mPSAD has been analysed in overall populations of men to enhance the detection of csPCa in relation to the PI-RADS score [19-22]. [I do not even think the word “overall” is needed or makes sense.
OK, changed: “mPSAD has been analysed in overall populations of men to enhance the detection of csPCa in relation to the PI-RADS score [19-22].”
80-83: this is too wordy. Shorten to: MRI-PMs are also attractive tools. However, few MRI-PMs have been developed from the latest versions of PI-RADS and external validation should be performed before their use. Easily accessible risk calculators (RC) are essential to avoid the cumbersome and often time-consuming nomograms [23-27].
Done: “MRI-PMs are also attractive tools, however, few of them have been developed from the latest PI-RADS versions, external validation is needed in each population where they are going to be implemented, and accessible risk calculators are essential to avoid the cumbersome and time-consuming nomograms [23-37].”
Note: I do not see any nomogram presented in paper by Mehralivand (ref. 26). I agree that the nomogram in the Truong paper (ref. 27) is onerous. I do not find the Lee nomogram very time-consuming or cumbersome. Also, all nomograms are able to be automated or converted into a computer calculator and made accessible on the internet. You may wish to note this in your discussion. An example of one such program is at http://www.prostatecalculator.org although this calculator does not relate to mp-MRI or mPSAD. Another excellent site is run by Tom Hueting of The Netherlands. See www.evidencio.com.
I completely agree with your comment. Mehralivand et al, developed their model but they do not construct any nomogram. www.evidencio.com is an excellent site. The fact is that today smartphone apps are the most practical way to apply nomograms as RCs. Thanks for your comment.
83: suggest distinguish the original Barcelona MRI-PM with the designation MRI-PMb (see annotation in margin for section 2.1 in the manuscript). I cannot tell from line 83 when you state “recent Barcelona MRI-PM if you mean the one already published or the one that is the subject of this paper. It appears that it is the original and already published one (ref. 38 ⇢ Morote J, Borque-Fernando A, Triquell M, Celma A, Regis L, Abascal JM, Sola C, Servian P, Escobar M, Mast R, et al. The Barcelona Predictive Model of Clinically Significant Prostate Cancer. Cancers. 2022, 14, 1589). It appears you do reference this on line 90. Perhaps moving that citation to line 86 would make it clearer to the reader which MRI-PM you are referring to.
I agree, and the sentence has been rewritten. “For the first time, the performance of an MRI-PM has been analysed with respect to PI-RADS categories, showing a net benefit over biopsy for all men in each PI-RADS <4 The designed RC also incorporates the novel option of selecting the csPCa probability threshold [38]. This option can be useful to facilitate future external validations [15] and outperformance the model in specific PI-RADS categories [39].”
92: clarify meaning of “and outperformance of the model in specific PI-RADS categories. Do you mean: and in the future to improve the performance of the model in specific PI-RADS categories.?
Yes, sentence has been changed: ”This option can be useful to facilitate future external validations [15] and improve the performance the model in specific PI-RADS categories [39].
96: Instead of “been analyzed after mp-MRI showing… consider “have improved the prognostic performance of mpMRI.”
and multiple markers combination have improved the prognostic performance of mpMRI [30].
98: Once again, a clearer statement is needed here.
The aim of this study is to compare the clinical usefulness of mPSAD with MRI-PMbdex for the proper….
OK, changed.
Section 2.1 (103-107): see annotations on manuscript.
104: incorrect English. Lines 103-105 would be better as: A head-to-head comparison of mPSAD vs. an adjusted MRI-PM (MRI-PMbdex) was designed from a multicenter trial conducted in the metropolitan area of Barcelona. The trial involved 2,432 men with suspected prostate cancer due to a serum PSA >3 ng/ml and/or an abnormal DRE.
Changed
105: The adjusted MRI-PM (MRI-PMbdex) was created from the development cohort of 1,486 men studied at the Vall d’Hebron Hospital (VHH) and the external validation cohort of 946 men from Parc de Salut Mar (PSM) and the Hospital Germans Trias I Pujol (GTiPH). That study was conducted between January 1 of 2006 and December 31 of 2019 [38].
Changed
129: no need for commas after radiologists or after institution.
OK
134-136: confusing as written; please clarify.
Sentence has been changed: “All men in the three participant institutions underwent 2 to 4-core mpMRI-TRUS cognitive-fusion guided biopsies to suspicious lesions and 12-core TRUS systematic biopsies when the PI-RADS reported in pre-biopsy mpMRI was 3 or high, while 12-core TRUS systematic biopsies when PI-RADS <3 [56].”
- Results
176-179: The DRE was abnormal in 25% of men; 28% had a prior negative biopsy and were scheduled for repeat biopsies, and 6.6% had a PCa family history. PCa was diagnosed in 1,214 men (49.9%), csPCa in 934 (38.4%) and iPCa in 280 (11.5%).
OK, PI-RADS information has been suppressed
Table 1. median PSAD is shown as ng/ml/cc. Density is weight per unit volume. The units should be ng/ml or ng/cc which is the same with PSAD. It should not be ng/ml/cc.
Sorry. ng/ml2
184: comma should be after the word categories.
OK.
186: change to: …except PCa family history, which showed a non-significant trend that was 4.7% with PI-RADS 1 and 8.3% for PI-RADS 5 (p =0.31).
Changed
Lines 189, 191-2: I believe you are looking at “frequency” rather than rate. : rate is defined as a quantity, amount, or degree of something measured per unit of something else. Other minor suggested changes shown.
Thanks, changed
193: suggest “Men with PI-RADS 4 were of a significantly higher age than those with PI-RADS 3, had a higher PSA and PSAD, a lower prostate volume, and more often had an abnormal DRE.”
Changed
195-197: again, it is frequencies rather than rates. Also clarify what PI-RADS groups you are referring to by changing (line 195) “were similar in PI-RADS 3 and 4 as was the frequency of iPCa. In the next sentence I do not know which PI-RADS category (3 or 4) had the higher frequency of csPCa. You need to clarify that. Other changes shown in annotations.
It is referred also to PI-RADS 3 and 4, it has annotated.
Line 199: suggest change to: Also, in the PI-RADS 5 category, the frequency of repeat biopsy was less as was that of iPCa, while the frequency of csPCa and overall PCa increased significantly (P <0.001).
Changed
208-212: see annotations in manuscript
Changed
Figures 1A and 1B: I think the legend is too small insofar as font size and will be difficult to read in the published article. Also, the legend for 1A shows PSAD and it is the mPSAD or MRI-PSAD, while the Model should be labeled as MRI-PMbdex or whatever you decide is the right abbreviation. The same is the case for the font size of the legend in 1B as well as the legend description. I honestly do not understand the item “Net Benefit Treat All”. The legend for the figure and the text in the manuscript refers to “biopsy all men”. If so, there should be consistency between text, legend and figure caption. In the legend, you do not expand the definition “DCA”.
Maybe you are right, for this reason I make the Figure bigger. Acronyms in the legend have been changed.
Decision curve analysis show the named “net benefit” over biopsy all men grey line according to the risk threshold (expressed in a continuous manner). Effectively, this is the reason why it is explained in statistic method.
Reviewer comment: I had to stop here due to sheer fatigue at reviewing this manuscript. I have spent far too many hours trying to understand what the writer(s) have said with a significant part of this time amending sentence structure and grammar. I believe I would need to spend another 4-6 hours to finish this review and I just do not have the time. I did show some additional editing as annotations in the manuscript.
I understand. I want to express my apologies and my thanks for your efforts. I send the manuscript for English Proofreading.
Kind regards
Juan Morote
Reviewer 2 Report
General comment
The manuscript entitled “Head-to-Head Comparison of Magnetic Resonance Imaging-Derived Prostate-Specific Antigen Density and a Predictive Model Based on Magnetic Resonance Imaging for the Proper Selection of Candidates for Prostate Biopsy in Men with Suspected Prostate Cancer.” aims to evaluate, via a head-to-head comparison the role of a predictive MRI model in the selection of candidates for prostate biopsy. The manuscript is well written, interesting and on point. Nevertheless, it requires a large revision in the discussion which is very limited compared to the importance of the findings reported. In addition, the limitations of the study are rushed and should be improved before considering the manuscript suitable for publication. In addition, it seems that some results should be double-checked. Finally, minor typos and English grammar throughout the manuscript should be revised as well.
- Major issues
TITLE
The title seems too long.
INTRODUCTION
76-77: regarding the issue of prostate volume, considering its importance in the PSA density.
MATERIALS AND METHODS
101: I don’t see any recall of the predictive model.
143: Two pathologists for each institution? Did you report a concordance analysis? If not, report this issue in the limitations of the study. This is also true for the radiologists involved.
RESULTS
Check data in the results. In particular, it seems that data reported related to figure 3 are the same as those reported to figure 4.
DISCUSSION
The discussion is very limited compared to the findings reported. This could heavily impact the overall quality of your study and should be mandatorily addressed. In addition, the discussion regarding predictive models should be also improved, considering the efforts in this direction of the recent literature. Please see, to this regard: doi: 10.3390/cancers12071767 and DOI: 10.3390/ijms22189971.
332-345: Albeit I appreciated the effort to critically discuss the results, in my opinion, this is not enough compared to the data reported. Please improve this section accordingly.
396-404: The limitations are far more numerous than those reported. Firstly, the study design is prone to sampling and selection biases. Secondly, there is no concordance analysis among pathologists, radiologists and urologists involved in the study. Albeit there is an inevitable inter variability among researchers involved in a multicentric study, this should be reported. Thirdly, both biopsies and PSA density suffers from intrinsic limitations related to the heterogeneity of prostate cancer as well as prostate volume. Finally, the lack of histopathological analysis of surgical specimens is another limitation. However, it is reasonable that not every patient underwent to radical prostatectomy.
- Minor issues
INTRODUCTION
63-65: Which scenarios?
MATERIALS AND METHODS
104: Specify why you choose a cut-off of PSA 3ng/ml and not 4ng/ml. Add references as well.
129: Add the years of experience of the radiologists involved.
136: As before, add years of experience of urologists involved.
149: Please explain what you mean by avoidable prostate biopsy.
Author Response
The manuscript entitled “Head-to-Head Comparison of Magnetic Resonance Imaging-Derived Prostate-Specific Antigen Density and a Predictive Model Based on Magnetic Resonance Imaging for the Proper Selection of Candidates for Prostate Biopsy in Men with Suspected Prostate Cancer.” aims to evaluate, via a head-to-head comparison the role of a predictive MRI model in the selection of candidates for prostate biopsy. The manuscript is well written, interesting and on point. Nevertheless, it requires a large revision in the discussion which is very limited compared to the importance of the findings reported. In addition, the limitations of the study are rushed and should be improved before considering the manuscript suitable for publication. In addition, it seems that some results should be double-checked. Finally, minor typos and English grammar throughout the manuscript should be revised as well.
Dear reviewer I thank your effort and apologise for the English. Proof-reading have been done by a specializsed company.
Kind regards
Juan Morote
- Major issues
TITLE
The title seems too long.
I agree and propose: “Comparison of PSA Density and an MRI-Based Predictive Model to Select Candidates for Prostate Biopsy”, lines 2-3.
INTRODUCTION
76-77: regarding the issue of prostate volume, considering its importance in the PSA density.
The paragraph in lines 73-76 has been modify: “PSAD is a classic tool improving the specificity of serum PSA [17] that is currently strengthened regarding the issue of prostate volume that is accurately provided in pre-biopsy MRI at no additional cost [18]. MRI-PSAD has been analysed in overall populations of men with suspected PCa and according to the PI-RADS categories [19-22]”.
MATERIALS AND METHODS
101: I don’t see any recall of the predictive model.
Paragraph in lines 100-105 has been rewrote: “A head-to-head comparison between MRI-PSAD and an adjusted Barcelona MRI-PM was carried out in a multicenter population of 2,432 men with PSA >3.0 ng/ml and/or abnormal digital rectal examination recruited in the metropolitan area of Barcelona. The adjusted Barcelona MRI-PM was formed from the development cohort (VHH, 1,486 men) and the external validation cohort (PSM and GTiPH, 946 men) recruited between January 1 of 2016 and December 31 of 2019 [38]
143: Two pathologists for each institution? Did you report a concordance analysis? If not, report this issue in the limitations of the study. This is also true for the radiologists involved.
One of two experienced uro-pathologist reported each biopsy in the three participant institutions. The paragraph in lines 140-143 has been rewrote: “Biopsy samples were sent separately to each pathology department where one expert uro-pathologists analysed biopsy specimens and reported the International Society of uro-pathology (ISUP) grade group (GG) when PCa was identified [57]. Complex cases were analysed by two expert uro-pathologist.
RESULTS
Check data in the results. In particular, it seems that data reported related to figure 3 are the same as those reported to figure 4.
Thanks. I have reviewed figure 3 and figure 4 and they correctly reported data of men with PI-RADS 3 and 4 respectively..
DISCUSSION
The discussion is very limited compared to the findings reported. This could heavily impact the overall quality of your study and should be mandatorily addressed. In addition, the discussion regarding predictive models should be also improved, considering the efforts in this direction of the recent literature. Please see, to this regard: doi: 10.3390/cancers12071767 and DOI: 10.3390/ijms22189971.
Discussion has been extended and suggested references have been incorporated
332-345: Albeit I appreciated the effort to critically discuss the results, in my opinion, this is not enough compared to the data reported. Please improve this section accordingly.
This paragraph has been improved according to your suggestion: “The characteristics of suspected PCa men with PI-RADS 3 were closer to the observed in men with PI-RADS 2 than to those of men with PI-RADS 4. In addition, the 16.9% csPCa detection rate of men with PI-RADS 3 was also closer to the 10.8% detected in men with PI-RADS 2 than the 52.2% observed in men with PI-RADS 4. Therefore, PI-RADS 3 is an uncertain scenario closer to negative MRI than PI-RADS 4, and this fact can justify some authors considering men with PI-RADS <4 as candidates to avoid prostate biopsy [9,15]. Finally, the characteristics of men with PI-RADS 5 are clearly different from those of men with PI-RADS 4. This fact, together with the increase of csPCa risk that reached 83.6% in men with PI-RAD 5 makes inadequate to considerer men with PI-RADS >3 in the same group, and data from with PI-RADS 4 and PI-RADS 5 should be considered separately [5]”.
396-404: The limitations are far more numerous than those reported. Firstly, the study design is prone to sampling and selection biases. Secondly, there is no concordance analysis among pathologists, radiologists and urologists involved in the study. Albeit there is an inevitable inter variability among researchers involved in a multicentric study, this should be reported. Thirdly, both biopsies and PSA density suffers from intrinsic limitations related to the heterogeneity of prostate cancer as well as prostate volume. Finally, the lack of histopathological analysis of surgical specimens is another limitation. However, it is reasonable that not every patient underwent to radical prostatectomy.
Thanks. All suggested imitations have been incorporated in the paragraph considering limitations of the study in lines 396-406: “Limitations of this study was the design itself prone to sampling and selection biases. There was not concordance analysis among pathologists, radiologists and urologists involved in the study. Inevitable inter variability among researchers involved in a multicentric study existed. Prostate biopsies and PSAD suffered from intrinsic limitations related to the heterogeneity of prostate cancer and prostate volume. The lack of histo-pathological analysis of surgical specimens is another limitation, although not every patient underwent to radical prostatectomy. Although the used definition of csPCa is the most spread, it may not represent the true csPCa. Because the adjusted Barcelona MRI-PM does not represent the developed model results cannot be attributed with certainty to the Barcelona MRI-PM. Further studies analysing the clinical usefulness of MRI-PSAD and MRI-PM according to PI-RADS categories are needed.”
- Minor issues
INTRODUCTION
63-65: Which scenarios?
“Nevertheless, uncertain scenarios in which high rate of unnecessary biopsies and overdetection of iPCa persist remain [8,9].”
MATERIALS AND METHODS
104: Specify why you choose a cut-off of PSA 3ng/ml and not 4ng/ml. Add references as well.
The threshold of 3 ng/ml is the currently recommended from the initial studies of ERSPC. References have been added, lines 99-102: “A head-to-head comparison between MRI-PSAD and an adjusted Barcelona MRI-PM was carried out in a multicenter population of 2,432 men with PSA >3.0 ng/ml [1,6] and/or abnormal digital rectal examination recruited in the metropolitan area of Barcelona.”
129: Add the years of experience of the radiologists involved.
It has been added, line 125: “An expert radiologist, with more than 5 year experience and more than 300 mpMRI per year”
136: As before, add years of experience of urologists involved.
It has been added in line 133-134: “one experienced urologist in each institution, with more than five year’s experience and more than 200 biopsies per year,”
149: Please explain what you mean by avoidable prostate biopsy.
Avoidable biopsies are those than can be avoided by using certain tools, due to its negative result or iPCa. I have substitute avoidable by avoided biopsies in the manuscript.
Additionally an English proofreading have been performed.
Thank you.
Juan Morote
Round 2
Reviewer 1 Report
Please see attachment.

Author Response
Dear reviewer,
I thank you for your comments and suggestions made on the manuscript
I have made all your suggested changes and below I make some comments
Table2
p Values in Table 2 have been verified and they are correct.
Las p Value was referenced to the comparison between all PI-RADS categories. However, I believe that is confuse, so I suppressed it.
Table 3-7
p value is referred to the significance of the distribution of men were sensitivity, specificity, NPV and PPV were estimated, and the odd ratio and 95% CI.
The grade groups are reflected to show if they are greater GG in higher PI-RADS. That means that csPCa detected would be more aggressive
Figure 5
The interpretation of DCAs is difficult because the net benefit is usually not calculated. The difference with the reference line of all men biopsied can be assessed for one specific threshold.
Reference 82 is a letter to the editor related to the reference 22. The comment we made was because our optimal PSAD threshold was different to that observed by Görtz in a similar series of men with PI-RADS 3
Respectfully
Juan Morote
Reviewer 2 Report
The authors improved the manuscript and no further corrections are required.
Author Response
We thank reviewer very much.